# Composite Analysis of the Virome and Bacteriome of HIV/HPV Co-Infected Women Reveals Proxies for Immunodeficiency

**DOI:** 10.3390/v11050422

**Published:** 2019-05-07

**Authors:** Juliana D. Siqueira, Gislaine Curty, Deng Xutao, Cristina B. Hofer, Elizabeth S. Machado, Héctor N. Seuánez, Marcelo A. Soares, Eric Delwart, Esmeralda A. Soares

**Affiliations:** 1Programa de Oncovirologia, Instituto Nacional de Câncer, Rio de Janeiro 20231-050, Brazil; sidoju@hotmail.com (J.D.S.); gcf.science@gmail.com (G.C.); esoares@inca.gov.br (E.A.S.); 2Vitalant Research Institute, San Francisco, CA 94118, USA; xdeng@vitalant.org (D.X.); delwarte@medicine.ucsf.edu (E.D.); 3Department of Laboratory Medicine, University of California San Francisco, San Francisco, CA 94143, USA; 4Instituto de Ginecologia, Universidade Federal do Rio de Janeiro, Rio de Janeiro 20211-340, Brazil; cbhofer@hucff.ufrj.br; 5Instituto de Puericultura e Pediatria Martagão Gesteira, Universidade Federal do Rio de Janeiro, Rio de Janeiro 21941-912, Brazil; emachado@infolink.com.br; 6Programa de Genética, Instituto Nacional de Câncer, Rio de Janeiro 20231-050, Brazil; hseuanez@inca.gov.br; 7Departamento de Genética, Universidade Federal do Rio de Janeiro, Rio de Janeiro 21944-970, Brazil

**Keywords:** virome, microbiome, HPV, HIV, immunodeficiency

## Abstract

The human cervical microbiome is complex, and its role in health and disease has just begun to be elucidated. In this study, 57 cervical swab samples from 19 HIV/HPV co-infected women were analyzed for both virome and bacteriome composition. Virome analysis focused on circular DNA viruses through rolling circle amplification followed by next-generation sequencing (NGS). Data were assigned to virus families and genera, and HPV types were identified. NGS data of bacterial 16S from a subset of 24 samples were assigned to operational taxonomic units and classified according to vaginal microbiome community state types (CSTs). Four viral families were found: *Papillomaviridae*, *Anelloviridae*, *Genomoviridae*, and *Herpesviridae*. Papillomavirus reads were more abundant in women with premalignant cervical lesions, which were also strongly associated with multiple (≥3) high-risk HPV infection. Anellovirus read abundance was negatively correlated with host CD4+ T-cell counts. The bacteriome revealed the presence of CST III and CST IV, and women with ≥1% frequency of genomovirus or herpesvirus reads displayed an increased risk of carrying CST IV. By characterizing the composition of the cervical circular DNA viruses and the bacteriome of HIV/HPV co-infected women, we identified putative interactions between these two microorganism communities and their associations with patients’ clinical characteristics, notably immunodeficiency status.

## 1. Introduction

The human microbiome consists of a collection of microorganisms such as viruses, bacteria, and fungi that symbiotically colonize the human body. This microbiome shows different profiles in each anatomical site of the body [1,2,3]. In the female anogenital tract, the viral and bacterial profiles are complex and present significant interpersonal variation [1,4,5,6]. In healthy women’s anogenital region, the *Papillomaviridae* is the most prevalent eukaryotic viral family [1,7]; its occurrence varies from 38% to 100% [1,5,7,8]. Anelloviruses are also frequent [5,9,10,11,12], while herpesvirus, polyomavirus, and adenovirus are usually detected in a small number of samples [1,5,7,13,14]. Among HIV-positive women, a similar diversity of viruses is found [4,9,15,16], except with a higher prevalence of papillomavirus infection [17,18,19,20,21,22,23,24,25].

Despite the recognized complexity of viral species in women’s cervical region, little is known about its association with bacterial communities in that anatomical site. There are five well-established distinct bacterial community state types (CSTs I–V) in the cervicovaginal region, classified according to their bacterial frequency profiles [26]. CSTs I, II, III, and V show a high frequency of *Lactobacillus* species (*L. crispatus*, *L. gasseri*, *L. iners*, and *L. jensenii*, respectively), while CST IV shows higher bacterial diversity and a high proportion of anaerobic bacteria such as *Prevotella*, *Dialister*, *Atopobium*, *Gardnerella*, *Megasphaera*, and *Sneathia* [26,27]. Interestingly, the CST IV interaction with HPV and HIV infection was shown in different studies [28,29,30]. Women with CST IV exhibited high proinflammatory cytokine production and CD4^+^ CCR5^+^ cell recruitment to the mucosal vaginal region, increasing the risk of HIV acquisition [28,29]. Moreover, the presence of CST IV has also been associated with a higher risk of HPV infection and cervical intraepithelial neoplasia [30,31,32,33,34]. On the other hand, most *Lactobacillus* sp. dominant CSTs (I, II, III, and V) show a negative association with HIV and HPV infections [28,29,30,35,36]. Less is known about the association and interaction of cervicovaginal CSTs with other viruses.

Research in the microbiome dramatically improved with the advent of novel sequencing methodologies. Together, these studies have exposed the complexity of the cervical virome and bacteriome in heath and disease. This is especially important in HIV-positive individuals, since HIV/AIDS has been related with enteric and blood microbiome alterations [37,38]. In this scenario, this study aims to characterize the cervical virome and bacteriome from HIV-positive women, to explore the interaction between these two communities and relate their patterns to the patients’ clinical characteristics.

## 2. Materials and Methods

### 2.1. Sample Selection

Cervical samples from 19 HIV/HPV co-infected women with multiple HPV infection were selected from a group of 140 HIV-positive women followed up between 2009–2011 through the “Program for HIV-infected Pregnant Women” in Rio de Janeiro, Brazil. These samples were assessed for HPV infection by PCR and Sanger sequencing and for risk factors for HPV infection and persistence in previous studies [39,40]. Patients selected for the current study had their cervical samples collected at the beginning of the second trimester of pregnancy (timepoint A), and at six (timepoint B) and 12 months (timepoint C) after delivery, and were positive for more than one type of HPV throughout the timepoints collected as determined by PCR and Sanger sequencing (either with multiple infections at least in one timepoint or carrying different HPV types at distinct timepoints). Therefore, a total of 57 samples were analyzed in this study. The study has been approved by the Ethics Research Committees of Universidade Federal do Rio de Janeiro (UFRJ) and of Instituto Nacional de Câncer (INCA) (reference protocols 029/08 and 142/10, respectively).

### 2.2. Circular DNA Enrichment and Sequencing

Cervical sample processing was performed as previously described [41]. Total DNA was extracted with the QIAamp DNA mini kit (QIAGEN, Valencia, CA, USA) followed by circular DNA enrichment by rolling circle amplification (RCA) with the Illustra TempliPhi Amplification kit (GE Healthcare Life Sciences, Piscataway, NJ, USA). Sequencing libraries were prepared using two nanograms of the purified RCA product and the Nextera XT DNA Sample Preparation kit (Illumina Inc., San Diego, CA, USA). Samples were indexed with distinct barcodes each and sequenced in an Illumina HiSeq 2500 platform (2 × 100 nt reads).

### 2.3. Virome Data Analysis

Reads obtained were sorted for each sample based on the dual barcodes used. Bacterial and human host reads were removed by mapping these sequences to the bacterial genomes present in the RefSeq database and the human reference genome hg19 using Bowtie 2 [42]. Duplicate reads (with identical bases from position 5 to 55 from the 5′ end) were identified, and only one was retained. Read nucleotides with Phred quality score <10 were trimmed. After this preprocessing, reads were submitted to a de novo assembly with Ensemble Assembler 1.0 [43] and assigned to the virus family and genus following a previously published methodology [44]. The contigs generated and single reads were submitted to a BLASTX [45] search against a virus protein database from RefSeq. Sequences that showed an *e*-value < 0.001 were compared to an in-house protein database with non-viral sequences derived from GenBank’s non-redundant database. Finally, the *e*-values were compared, and sequences that showed the lowest *e*-value when compared to virus protein sequences were classified into a virus family according to the most similar sequence in the database.

To avoid false-positive results, virus families with less than 50 reads were excluded from further analyses. During the removal of host reads, human endogenous retroviruses (HERV) were also discarded, since they represent greater than 8% of the human genome [46]. As most of the reads attributed to *Retroviridae* were HERVs, we also excluded this family from further analysis to avoid the biased underrepresentation of this virus family. We have used the normalized number of reads to each virus family in our analyses. The number of reads assigned to each virus family from the remaining viruses was normalized by calculating the log10 of 1 plus the number of reads assigned to each viral family divided by the total number of reads multiplied by one million (log10 1 + (number of reads assigned to each genus or viral family/total number of reads × 1,000,000)). We have also normalized the number of reads assigned to each virus family divided by the sequences submitted to BLASTX (reads of viral origin). Finally, the proportion of reads of each virus family was also used, calculated by dividing the number of reads assigned to each virus family by the total number of sequences submitted to BLASTX.

### 2.4. Papillomaviridae Identification and Diversity

Virome reads from each sample were mapped to genomes of different HPV types with BWA [47]. Reads that mapped to multiple HPV references or regions were randomly assigned to one region/reference. The number of reads mapped to each reference was measured with SAMTOOLS [48] as described previously [41]. The percentage of reads of each HPV type was calculated as a fraction of the total reads mapped to any HPV reference genome used.

### 2.5. Bacterial Community State Type (CST) Characterization

The bacteriome analyses were performed for the postpartum samples (timepoints B and C) from 12 out of the 19 women of this study, because their bacterial 16S gene had already been sequenced in a previous study [49]. Briefly, a 724-bp fragment spanning the variable regions V3 to V6 of the bacterial 16S rRNA gene was PCR-amplified using primers 338F and 1061 [49]. The 16S sequence data were reanalyzed herein to evaluate the association of bacterial CSTs with the cervical virome. Briefly, the reads from bacterial 16S rRNA sequences were assigned to operational taxonomic units (OTUs) using QIIME’s closed reference script (Greengenes Database). OTUs with relative abundance lower than 0.005% were discarded, and those remaining were summarized at the species level. Unsupervised hierarchical clustering based on Bray–Curtis dissimilarity and average linkage was performed in the *R* environment and was applied to define clusters according to the abundance and taxa diversity of each sample. The clusters found were assigned to vaginal microbiome CSTs, as defined in previous studies [26,50].

### 2.6. Genomoviridae Complete Genomes

Samples with reads from *Genomoviridae* were submitted to an assembly with reference using Geneious^®^ v.9.1 (Biomatters, Auckland, New Zealand). The reference used for each sample was the sequence from the database that showed the best similarity score to the reads in BLASTX searches. The consensus sequences were extracted and aligned using ClustalW2 [51]. Phylogenetic analysis was performed with PhyML 3.0 [52] with the most fit evolutionary model determined by Model Generator [53].

### 2.7. Assessing the Presence of Fungus in the Samples

Since the only genomovirus with known tropism infects fungi [54], four cervical samples with reads to this viral family and six samples without reads of *Genomoviridae* were analyzed for the presence of fungal DNA using semi-nested PCR with ITS1/ITS4 primers in the first round and ITS86/ITS4 in the second round under previously described conditions [55], except for the enzyme used, which was Taq DNA polymerase (New England Biolabs, Ipswich, MA, USA). PCR-amplified products were detected in 1% agarose gel electrophoresis and purified with a QIAquick PCR purification kit (QIAGEN) following manufacturer’s instructions. Purified products were sequenced by Sanger with the primers used in the second PCR round, and the sequences generated were assembled and edited with Geneious^®^ v.9.1 (Biomatters). The consensus sequences were extracted and submitted to BLASTN to identify the fungi present in the samples.

### 2.8. Statistical Analyses

The frequency of reads or the normalized number of reads of each virus family was used to calculate the Spearman’s correlation coefficient with the HIV viral load and CD4^+^ T-cell counts. The Mann–Whitney *U* test was used to compare the difference of read distribution or normalized number of reads for all virus families between samples with distinct cytological results. The odds ratio (OR) and 95% confidence interval values were estimated for the association of cytological lesions with multiple infections by high-risk HPV types. Similarly, the comparison of the frequency of reads or normalized number of reads of each virus family to CSTs was determined with the Mann–Whitney test. The association and risk analysis of CSTs with the presence of HPV16 was performed with the relative risk (RR) statistics. The RR was also used to analyze the association of frequency below 1% of *Anelloviridae*, *Genomoviridae*, or *Herpesviridae* with CSTs. All statistical analyses were performed with SPSS (IBM Corporation, Chicago, IL, USA), and graphical representations were generated with GraphPad Prism (GraphPad Software Inc., San Diego, CA, USA) or with Microsoft Excel (Microsoft Corp., Seattle, WA, USA).

### 2.9. Data Availability

The consensus sequences obtained in this study were deposited in Genbank under the accession numbers MK513442 and MK513443. Sequencing reads were submitted to the Sequence Read Archive (SRA) under the project numbers PRJNA418043 and PRJNA392046.

## 3. Results

The 19 patients studied herein were selected from a cohort previously analyzed for HPV infection, persistence, and associated risk factors [39,40]. The average age of the patients was 28 years, and 53% were married/co-habited with a single partner (Table 1). Eight patients were current smokers at the sample collection time or smoked previously, six had less than four lifetime sexual partners, and 10 had a previous history of sexually transmitted diseases other than HIV infection. Only four women had already been under antiretroviral therapy (ART) before conception, and all of them had been treated with combinatorial ART during pregnancy. The median CD4^+^ T-cell counts were 327/mm^3^, and the HIV-1 viral load was 8227 copies/mL at moment of the study’s enrollment (timepoint A). At the first, second, and third collection timepoints, nine, 14, and 10 women presented low-grade or high-grade squamous intraepithelial lesions (LSIL or HSIL), respectively. During the complete follow-up, 16 women presented LSIL or HSIL at least in one timepoint studied (Table 1).

The average number of reads obtained per sample after trimming out the low-quality reads was 4,193,239 (931,784–18,616,732). Overall, four viral families presented more than 50 reads in the samples. The most representative was *Papillomaviridae*, with 86% of the reads, followed by *Anelloviridae* (12%), *Genomoviridae* (2%), and *Herpesviridae* (0.06%). Papillomavirus reads were found in all but two of the 57 samples (44A and 44B), and reads from *Genomoviridae* were found in a high proportion in three samples (Figure 1). The normalized number of reads from papillomaviruses was significantly higher (*p* < 0.001) in women that presented any type of intraepithelial lesion (LSIL or HSIL) compared to women with normal or atypical squamous cells of undetermined significance (ASCUS) cytology (Figure 2A). A correlation analysis of clinical HIV-related data (available only for the first collection timepoint) to the normalized number of anellovirus reads per total reads showed a moderate but significant negative correlation with the CD4^+^ T-cell counts of the subjects (r_s_ = −0.499; *p* = 0.03) (Figure 2B). The normalized number of anellovirus reads per viral reads showed similar significance (r_s_ = −0.490; *p* = 0.033), as well as the percentage of anellovirus reads (r_s_ = −0.525; *p* = 0.021). The association between HIV viral load and anellovirus reads was not significant in any read normalization calculated (*p* = 0.099, *p* = 0.056, and *p* = 0.259, respectively).

Reads from two samples (30A and 87B) assembled one complete genomovirus genome each. Both genomes had the three characteristic open reading frames (replication associated protein, or Rep; RepA, and CP) and the nonamer of the stem-loop structure (Figure 3A,B). A phylogenetic analysis using the Rep amino acid sequences showed that the two viruses found were different species of the *Gemykibivirus* genus (Figure 3C). The gemykibivirus 30A (GmkV30A) clustered with the *Human-associated gemykibivirus 1* species includes viruses isolated from human blood (acc.# LK931485) [56], plasma (acc.# KP974694) [57] and sewage (acc.# KJ547644 and acc.# KJ547645) [58]. The GmkV87B, on the other hand, clustered with the *Human-associated gemykibivirus 2* species, which includes strains isolated from human cerebrospinal fluid (acc.# KP133075, acc.# KP133076, and acc.# KP133077), from human feces (acc.# KP133078 and acc.# 1330799) and from sewage (acc.# KP133080) [59].

The only genomovirus with a confirmed host is the *Sclerotinia sclerotiorum* hypovirulence-associated DNA virus 1 (SsHADV-1) that infects the plant pathogenic fungus *Sclerotinia sclerotiorum* [54]. Considering this and the ubiquous presence of members of the *Genomoviridae* family, we evaluated the presence of fungi DNA in a subset of cervicovaginal samples. Samples 30A, 31A, 44A, and 87B, presenting *Genomoviridae* reads, and samples 4A, 48A, 58A, 67A, 101B, and 118A, with no evidence of *Genomoviridae* reads (both sets chosen randomly, but including the two samples from which the complete genomes were derived), were analyzed by PCR. *Candida* sp. was identified in three samples (4A, 30A, and 31A), and *Sarocladium* sp. was identified in one sample (87B). Therefore, fungal DNA was identified in three (30A, 31A, and 87B) of four samples with *Genomoviridae* reads and in one of the controls.

When analyzing the reads assigned to different HPV types and assembling their genomes, we were able to identify 41 different HPV types belonging to the *Alphapapillomavirus* genus. Patients showed different HPV type proportion patterns among them, and there was also intrapatient HPV type variation over time (Figure 4). The most prevalent HPV type was HPV16 (present in 79% of the patients in at least one timepoint), followed by HPV51 and HPV56 (53% each), HPV52 (47%), and HPV59 (37%). Four HPV types were found in only one sample.

Reads of multiple HPV types were found in 49 (89%) samples (Figure 4). An average of five different HPV types was found per sample, varying from one to 20 (sample 83A). More than one high-risk or probable/possible high-risk HPV (hr-HPV) types were observed in 87% of the samples. The two samples displaying HSIL (32C and 103B) had reads of multiple HPV types including HPV16 and other high-risk (hr) types (Figure 4). Multiple infections by three or more hr-HPV types were strongly associated with the presence of LSIL or HSIL (odds ratio 3.2, 95% CI 1.1–9.7).

When different timepoints of the same patient are compared for HPV type persistence, HPV33 was found in all three follow-up samples of the two women positive for this virus type (Figure 4). HPV16 was found in 15 women, and persisted in all three timepoints in nine (60%) of them. The HPV types that showed the greatest rates of persistence in at least two collection timepoints were HPV33 and HPV73 (100% persistence), HPV67 (75%), HPV16 (73%), and HPV59 (71%). Twelve HPV types were present in only one collection timepoint, and four of them were present only in the first collection timepoint, i.e., the second trimester of pregnancy (HPV40, *n* = 2; HPV84, *n* = 1; HPV86, *n* = 3; HPV102, *n* = 2).

Although samples with HSIL have a variety of HPV types (32C, *n* = 4; 103C, *n* = 12), most of their reads (>96%) are assigned to only one (103C) or two (32C) HPV types (Figure 4). The percentage of viral reads from high-risk or probable/possible hr-HPV types was greater than those of other types in almost all the samples, regardless of cytological results (39/55 samples, 71%). 

The hierarchical clustering analysis of the bacteriome data revealed the presence of two distinct CSTs: CST III (*Lactobacillus iners*-dominant) and CST IV (high proportion of anaerobic bacteria). They were present in 32% and 68% of the samples, respectively (Figure 5). When samples classified with distinct CSTs were compared for the presence of HPV16 (the most frequent HPV found), those harboring CST IV displayed a higher proportion of infections without HPV16 (33%) than those with CST III (14%) (Figure 6), but the risk association between CST IV and HPV16 was not statistically significant (RR = 0.75 (0.44–1.26)).

We further analyzed the association of bacterial CSTs to the relative frequency or the normalized number of reads of all virus families found, except for *Papillomaviridae* (which was present in virtually all samples), but no significant differences were found (Figure 7A–C). However, the categorical analysis of viral frequency showed an association of genomovirus and herpesvirus with CST IV (Table 2). Women with frequency ≥1% of genomovirus and herpesvirus reads displayed an increased risk of carrying CST IV, (RR = 1.54 (1.11–2.12), and 1.47 (1.10–1.95), respectively).

## 4. Discussion

The human microbiome is complex, and its role in health and disease is only beginning to be elucidated. While many recent studies have been conducted to unravel the interaction of the human virome [1,4,7,60,61,62] and bacteriome [3,26,32] with their host, the interplay between these two microbiome components is scarcely addressed. In the present study, we characterized the main DNA viruses as well the bacterial components present in the HIV-positive women’s cervical region through next-generation sequencing, allowing us to address their mutual relationships. Moreover, this analysis allowed us to associate patterns of these two microorganism communities with the clinical characteristics of the patients.

The majority of viral reads found in the patients’ samples analyzed herein were assigned to *Papillomaviridae*. This was especially true for samples of women displaying high-grade or low-grade intraepithelial lesions. Although our samples were previously selected for multiple HPV infection, unbiased cervical virome studies have found papillomaviruses as the most prevalent eukaryotic infecting virus irrespective of the genital tract health [1,4,7,8]. It is worth mentioning that the rolling circle amplification (RCA) method that was used favors the detection of circular DNA viruses (including anelloviruses and HPV), and may be selecting against the detection of other viruses, such as herpesviruses, which was a limitation of the current study. Besides the inter-individual diversity previously observed in cervical viromes [1], our findings also highlight the importance of HPV read counts, which could be a proxy of the HPV viral load [4,63] in the presence of cervical abnormalities [64,65,66].

Anelloviruses are frequently found in the cervical region regardless of the presence of abnormalities [9,10,11,12,67,68]. However, they are more prevalent in HIV-positive patients [69], and viral load in the blood is higher in immunosuppressed individuals [70,71] and increase with AIDS progression [37,72], suggesting a negative association with the host immune response. Some studies propose the use of anellovirus viral load in blood as a marker to monitor immunologic reconstitution in immunossupressed and in HIV-positive patients [37,71,73,74]. Our study shows that such association between the immune response and anellovirus normalized reads or read proportion can be extended to the cervical region, showing that anellovirus read counts in this mucosa are inversely correlated with CD4^+^ T-cell counts in the periphery.

*Genomoviridae* is a recently established viral family [75] that includes the only identified ssDNA virus that infects fungi [54]. More than 100 members of this viral family have been described from different isolation sources (plants, animals, and environmental) [75,76]. Human-associated gemykibiviruses have been previously described from human pericardial [77] and cerebrospinal [59,78] fluids, plasma [79,80], blood [56], feces [59], and also from sewage [9,58]. In this study, we characterized two complete genomes of this virus identified in the female genital region. Other metagenomics studies investigating cervical samples [1,4,7] did not find this viral family, which was probably because it had only recently been described in humans [56,57,59,77,78,80]. Even though one of the four genomovirus-positive samples analyzed in this study was negative for fungal DNA, the prevalence of fungal DNA-positive samples was higher among the genomovirus-positive samples (75%) than in the negative counterparts (17%). Novel studies evaluating genomovirus replication or seroconversion will be needed to identify the host of different genomoviruses.

Our analysis of samples previously identified as HPV-positive by Sanger sequencing failed to identify papillomavirus reads in two samples from the same patient (44A and 44B). These results may be due to the limitations of the methodology used, since RCA will amplify circular DNA, and consequently will fail to detect HPV copies integrated into the host DNA, which may be the case for those samples. Despite our limitations, we could observe a high inter-patient variability regarding the frequency and diversity of different HPV types. Concomitant infection with more than one HPV type was associated with an increased risk of cervical lesions in several studies [81,82,83,84] and with HPV persistence [85]. In our study, we found a higher risk for the presence of cervical abnormalities in women that carried at least three high-risk or probable/possible high-risk HPV types. Also in agreement with previous reports, we found a higher prevalence of HPV persistence with the high-risk HPV types [86,87,88]. We could not evaluate the overall risk of HPV multiple infections or the impact of HPV persistence, because our samples were selected based on the identification of more than one HPV over three collection timepoints.

The bacteriome has been recently described as a critical factor that could affect the increase or decrease of viral infections [89]. The female lower reproductive tract harbors complex communities of microorganisms that are important to maintain health and protect the cervix from viral infections. Bacterial cervix communities are clustered into five groups named community state types (CST) I to V. CSTs I, II, III, and V are predominantly formed by *Lactobacillus* sp. (*L. crispatus*, *L. iners*, *L. gasseri*, and *L. jensenii*, respectively). CST IV exhibits higher species diversity and a high proportion of multiple anaerobic bacteria, including *Gardenerella*, *Prevotella*, *Atopobium*, *Sneathia*, *Aerococcus*, and *Megasphaera* [26,50]. In the current study, we showed the presence of only two CSTs (III and IV) in the samples analyzed. Recent studies have shown that CSTs III and IV are associated with a higher prevalence of HIV, HPV, and HSV-2 infections [35,90]. However, little is known about the relationship of bacterial CSTs with others viral infections. We analyzed CST association for three viral families (*Anelloviridae*, *Herpesviridae*, and *Genomoviridae*) found in this study, which had not been previously analyzed. We observed a higher frequency of samples with read abundance ≥1% of *Genomoviridae* and *Herpesviridae* in CST IV, showing an increased risk of 54% and 47%, respectively. Additionally, CST IV samples also displayed a higher proportion of HPV non-16 viruses. The mechanisms involved in the establishment of such viral infections and how the bacteriome modulates or is modulated by them remains to be elucidated. However, our data strongly point out to a scenario in which a more immunodeficient status, which is commonly associated with CST IV, also allows the establishment of viral infections that are less frequently detected in immunocompetent hosts. This is the case of genomoviruses (associated with fungal co-infections), herpesviruses, and HPV non-16 types, as seen herein.

Despite the small number of patients followed herein, this study was able to describe the composition, dynamics, and the putative interaction of the cervical circular DNA virome and the bacteriome in a highly susceptible group of HIV-positive pregnant women. However, additional studies are necessary to clarify the role and the mechanisms of the microbiome homeostasis and its association with the host health and disease, including the development of cervical lesions and cancer.

## Figures and Tables

**Figure 1 viruses-11-00422-f001:**
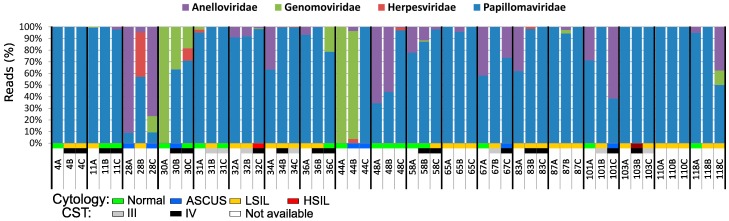
Distribution of the reads assigned to each virus family found in the cervical samples over time. A number was assigned to each patient studied followed by letters that represent the collection timepoints studied. The cytological results (normal, ASCUS, LSIL, or HSIL) and the CSTs of the samples are color-coded according to the graphical legend at the bottom of the Figure. A: beginning of second trimester of pregnancy; B: six months after delivery; C: twelve months after delivery; ASCUS: atypical squamous cells of undetermined significance; LSIL: low-grade squamous intraepithelial lesion; HSIL: high-grade squamous intraepithelial lesion; CST: community state type.

**Figure 2 viruses-11-00422-f002:**
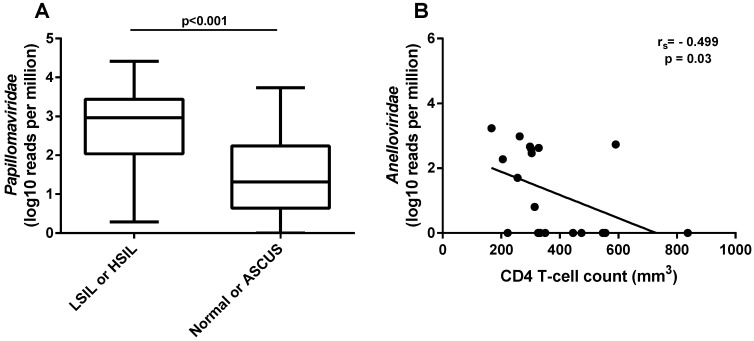
Clinical data and normalized number of reads from virus families. (**A**) Comparison of normalized number of viral reads assigned to *Papillomaviridae* to total number of reads between samples with normal or ASCUS (atypical squamous cells of undetermined significance) and with LSIL or HSIL (low-grade or high-grade squamous intraepithelial lesion, respectively). (**B**) Negative correlation between the normalized number of reads assigned to *Anelloviridae* to total number of reads and patients’ CD4^+^ T-cell counts per mm^3^ of blood at the time of sample collection.

**Figure 3 viruses-11-00422-f003:**
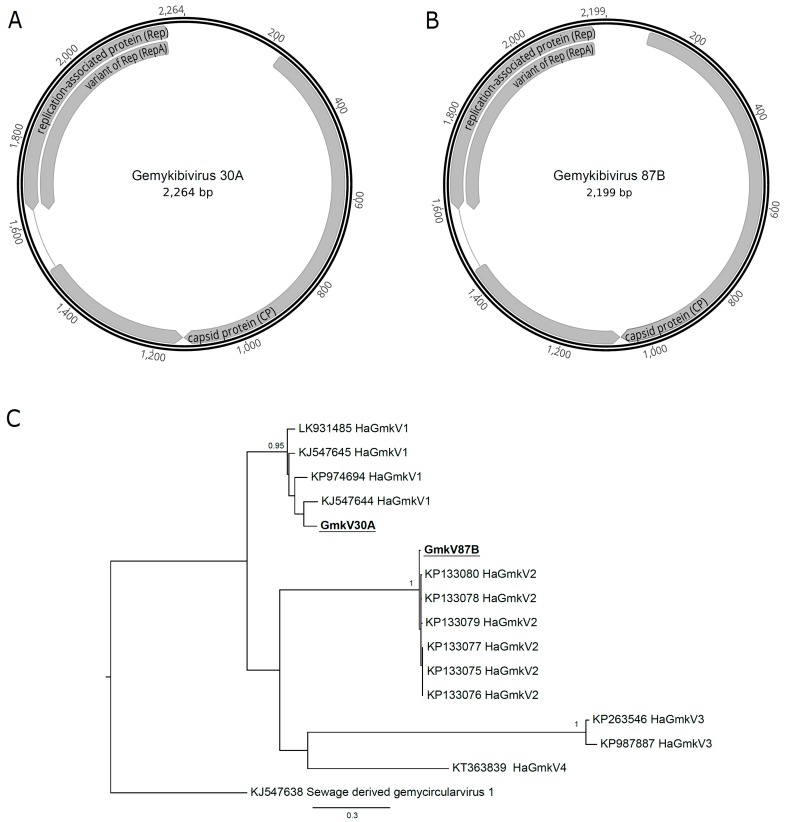
*Human-associated gemykibivirus* (HaGmkV) genome sequences and phylogenies. Genome organization of gemykibivirus 30A (**A**) and 87B (**B**), isolated from the respective samples and collection timepoints. (**C**) Maximum likelihood phylogenetic tree reconstructed with amino acid sequences from the gemykibivirus replication-associated protein. Phylogenetic analysis was performed with PhyML 3.0 [52] with the evolutionary model determined by the Model Generator [53] and 1000 bootstrap replicates. Only bootstrap values above 70% are shown in the tree. Sequences assembled in this study are bolded and underlined. Sequences retrieved from Genbank are identified with their respective accession numbers.

**Figure 4 viruses-11-00422-f004:**
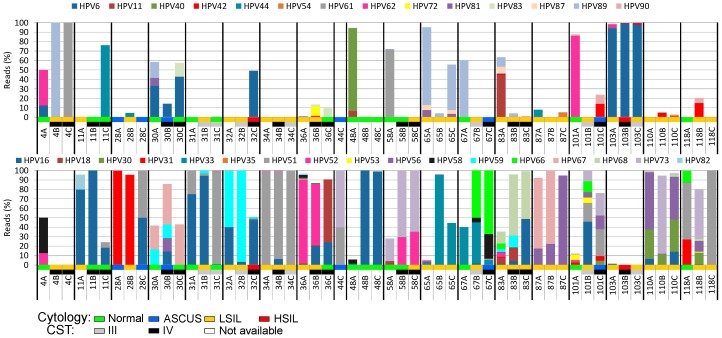
Cumulative distribution of reads mapped to each HPV type found in the analyzed cervical samples over time. A number was assigned to each patient studied followed by letters that represent the collection timepoints studied. The cytological results (normal, ASCUS, LSIL, or HSIL) and the CSTs of the samples are color-coded according to the graphical legend at the bottom of the figure. The upper panel depicts low-risk HPV types, while the lower panel shows the high-risk HPV types found. A: beginning of the second trimester of pregnancy; B: six months after delivery; C: 12 months after delivery; ASCUS: atypical squamous cells of undetermined significance; LSIL: low-grade squamous intraepithelial lesion; HSIL: high-grade squamous intraepithelial lesion; CST: community state type.

**Figure 5 viruses-11-00422-f005:**
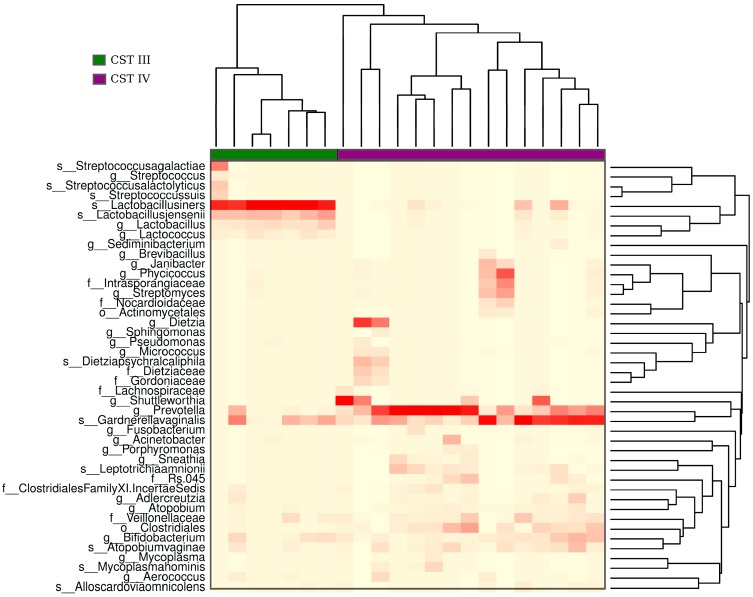
Heatmap of hierarchical clusterization of the bacteriome present in the analyzed samples. The heatmap shows the abundance of bacteria within the samples. The color gradient of red to white in the heatmap represents the gradient of high to low abundance of bacterial species, respectively. Based on the abundance of bacteria, samples were clustered into two groups: CST (community state type) III (green) and IV (purple). CST III exhibits a high abundance of *Lactobacillus iners* and CST IV shows a mix of bacterial species, with a high abundance of *Prevotella* and *Gardenerella*.

**Figure 6 viruses-11-00422-f006:**
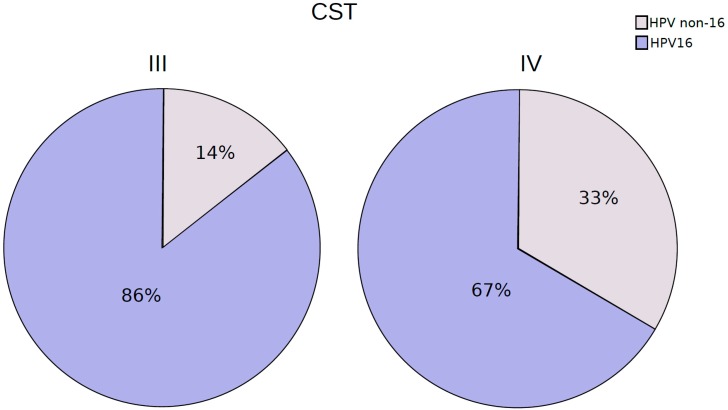
Proportion of patients harboring HPV-16 reads among CST III and IV cervicovaginal samples.

**Figure 7 viruses-11-00422-f007:**
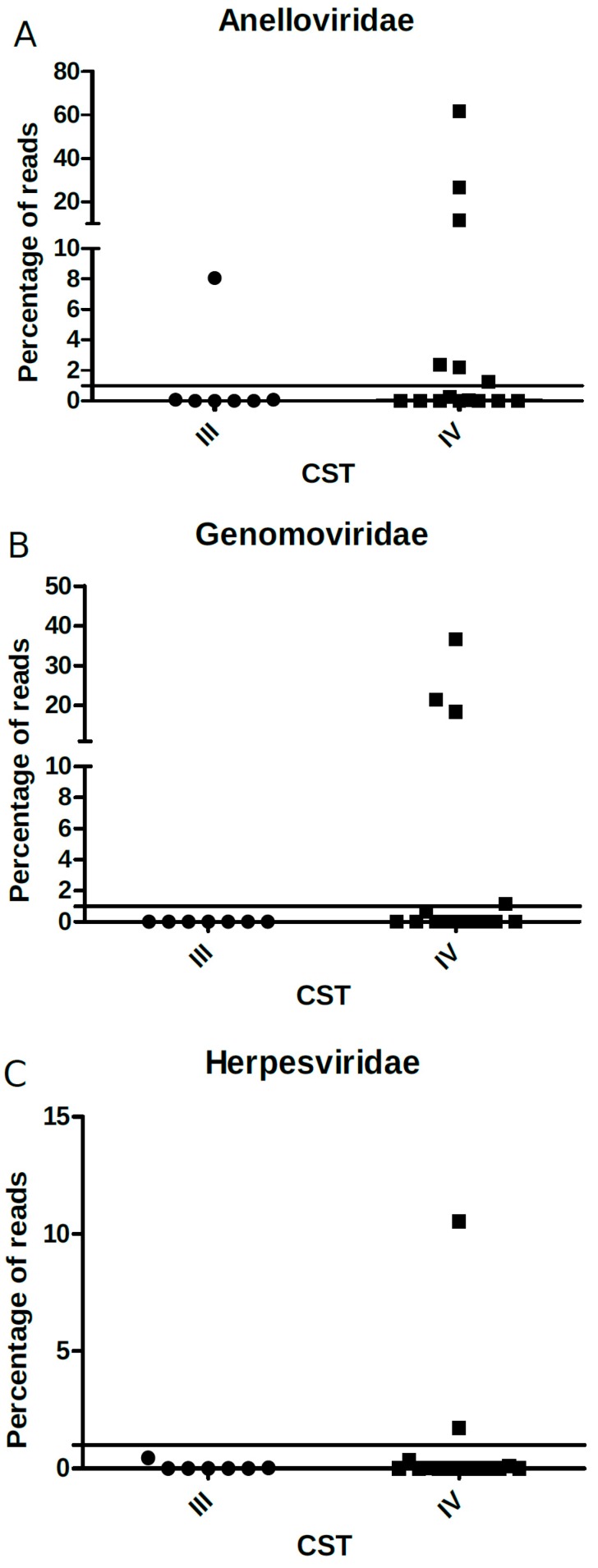
Percentage of (**A**) *Anelloviridae*, (**B**) *Genomaviridae*, and (**C**) *Herpesviridae* viral reads (*y*-axis) relative to each CST type (*x*-axis). The solid black line depicts the proportion of 1% in the *y*-axis and each point within the graph represents one sample analyzed (circles: CST III; squares: CST IV).

**Table 1 viruses-11-00422-t001:** Demographic and clinical characteristics of the 19 HIV/HPV co-infected women studied.

Characteristic	*n*	%
Average age (years; ± SD)	28 ± 6	
Married/co-habitating status	10	53
Past/Present smoking	8	42
Number of sexual partners ≥ 4	13	68
Previous sexually transmitted infection	10	53
ART initiation before conception	4	21
Median CD4^+^ T-cell counts at study enrollment (cells/mm^3^; IQR_50_)	327 (289–492)	
Median HIV viral load at study enrollment (copies/mL; IQR_50_)	8227 (933–13,824)	
**Presence of high-grade or low-grade squamous intraepithelial lesion**		
First collection	9	47
Second collection	14	74
Third collection	10	53

ART, antiretroviral therapy; IQR_50_, interquartile range 50%.

**Table 2 viruses-11-00422-t002:** Relative risk for CST IV associated with high read frequency (≥1%) of *Herpesviridae*, *Genomoviridae*, and *Anelloviridae* among HIV/HPV-positive women.

Virus Family	Read Frequency	CST % (N/Total)		Relative Risk	95% CI
		III (*N* = 7)	IV (*N* = 17)		
*Herpesviridae*	<1% (*N* = 22)	32 (7/22)	68 (15/22)	1.47	1.10–1.95
	≥1% (*N* = 2)	0 (0/2)	100 (2/2)		
*Genomoviridae*	<1% (*N* = 20)	35 (7/20)	65 (13/20)	1.54	1.11–2.12
	≥1% (*N* = 4)	0 (0/4)	100 (4/4)		
*Anelloviridae*	<1% (*N* = 17)	35 (6/17)	65 (11/17)	1.32	0.83–2.10
	≥1% (*N* = 7)	14 (1/7)	86 (6/7)		

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
