# Peer review of "Composite Analysis of the Virome and Bacteriome of HIV/HPV Co-Infected Women Reveals Proxies for Immunodeficiency"

_viruses, 2019, doi:10.3390/v11050422_

Round 1
Reviewer 1 Report
This is a well-written paper describing the viruses and bacteria in cervical samples from 19 HIV/HPV-coinfected women. The data are presented clearly, and the data shown support the conclusions.
There are some points that should be clarified prior to publication:
-Can the authors provide additional evidence that the genomovirus is not a contaminant?
- Read abundance is used as a metric to indicate virus abundance. However, the amplification steps used in sample preparation skew the abundance of the viruses. If conclusions based on quantitative data are included, the authors should accurately quantitate the viruses (possibly by pcr). This applies to analyses throughout the paper.
-Can the authors explain why the frequency of 1% of genomovirus and herpesvirus was used as a cutoff? Why not just consider positivity? Due to amplification, it seems this percentage is rather arbitrary. Do the data change if positivity is used instead?
- Were there any consistent differences in the samples collected during pregnancy compared to those after pregnancy that might be pregnancy-related effects?
-How were duplicate reads identified?
-Why were virus families with<50 reads excluded? How was this number determined? 50 reads is still a significant signal, and many true positives are likely being excluded. Does the inclusion of other viruses change the results?
-Were there individual sequences that mapped to multiple HPV types when typing the viruses? How were these resolved? (Section 2.4)
-Although the bacterial data has previously been published, it would be useful here to at least give more data regarding the amplicon sequencing strategy (full length? Which variable regions if shorter?)
-Line 143 typo in polymerase
- Figure 1 - the inclusion of the cytology on this figure is very helpful. The labels are a bit hard to read, though, so maybe indicate the types with a shape or color block above or below
-Figure 5 - it would be useful to see the viral data represented on this figure as well. At least at the family level.
-It should be noted in the discussion that the amplification, because it favors circular DNA, is an advantage for detecting some viruses (anello, papilloma,...) but may be selecting against detection of other viruses in the vagina (EBV, CMV, HSV (although some were detected, there may've been more), molluscum contagiosum, etc.).
Minor points:
The title indicates the paper is about the community composition in pregnant women, but only 1/3 time points is in a pregnant woman. This should be clarified. Also this title does not truly reflect the content of the paper.
Very nice work, overall.
Author Response
Dear Dr. Tachedjian,
Please find enclosed the revision to our manuscript entitled “Composite Analysis of the Virome and Bacteriome of HIV/HPV-Coinfected Pregnant Women Reveals Proxies for Immunodeficiency”, to be considered for publication in Viruses as an Article.
As you will see below, we have incorporated/modified all suggestions by the two Reviewers to our manuscript. Each individual criticism has been addressed, and all changes to the manuscript have been highlighted in yellow in the revised version of the text. All authors have read the revised version of the manuscript and agreed with this re-submission.
REVIEWER 1
This is a well-written paper describing the viruses and bacteria in cervical samples from 19 HIV/HPV-coinfected women. The data are presented clearly, and the data shown support the conclusions.
There are some points that should be clarified prior to publication:
Can the authors provide additional evidence that the genomovirus is not a contaminant?
Despite the fact that we did not try to directly identify genomoviruses in the samples by PCR or real-time PCR, a number of evidences suggest these were not contaminants: 1) We do not carry this virus or ever worked with it in the laboratory where all experiments were carried out; 2) All DNA circome samples were run together (in the same run experiment), and only a few samples were positive for genomoviruses; 3) Although not completely concordant (100%), we had a high concordance between the detection of genomoviruses and their alleged hosts (fungi). The presence of fungi was tested by PCR and 3 out of 4 genomovirus-positive samples had fungal DNA detected, whereas only one genomovirus-negative sample out of 3 tested was positive for fungal DNA. Therefore, we think those viruses did not constitute sample contaminants.
Read abundance is used as a metric to indicate virus abundance. However, the amplification steps used in sample preparation skew the abundance of the viruses. If conclusions based on quantitative data are included, the authors should accurately quantitate the viruses (possibly by pcr). This applies to analyses throughout the paper.
The rolling circle amplification (RCA) step used to amplify the signal of circular DNA viruses in the samples uses hexameric random primers which are highly nonspecific, so we do not anticipate any bias among the samples analyzed in that regard. The identification of the viruses found is done at the level of virus family (the lowest reliable taxonomic level of identification), and the use of specific PCR primers for their deeper identification is not possible. Moreover, designing primers based on consensus sequences representing virus families would rather lead to the same technical bias problem as raised by the Reviewer during their PCR amplification.
-Can the authors explain why the frequency of 1% of genomovirus and herpesvirus was used as a cutoff? Why not just consider positivity? Due to amplification, it seems this percentage is rather arbitrary. Do the data change if positivity is used instead?
We used the arbitrary cutoff of 1% relative abundance based on multiple reports of microbiome studies that conducted unsupervised clustering analyses and that used such cutoff value as a proxy for data with potential biological relevance. The highest number of reads for a given virus family representing less than 1% of the reads in a sample was 47, followed by 8 reads (the second highest). As per the Reviewer´s request, however, we have conducted the same statistical analyses using crude positivity (the presence of 1 or more read), even realizing that we would likely be including several false positive samples (with very few numbers of reads) as true positives (this information was added to the revised version of the ms; lines 114-19 and changes to sentences in lines 158, 160-1, 164). When doing this, the genomovirus and herpesvirus associations with CST IV loose significance. Therefore, we truly think the relative abundance, rather than the absolute positivity, is much more likely to represent the actual biological distribution of the samples. The analysis using read abundance instead of absolute read counts is also more conservative to find potential associations.
Were there any consistent differences in the samples collected during pregnancy compared to those after pregnancy that might be pregnancy-related effects?
We have not carried out this analysis, since the viral profiles of the three time points from each patient did not vary substantially (e.g., see Figure 1). Such analysis would likely fail to tease out any significant differences.
-How were duplicate reads identified?
Duplicate reads were identified as reads containing identical bases from position 5 to 55 from the 5' end. We added this information to the revised manuscript (line 99).
-Why were virus families with<50 reads excluded? How was this number determined? 50 reads is still a significant signal, and many true positives are likely being excluded. Does the inclusion of other viruses change the results?
The “<50 reads” cutoff used was based on the overall sum of reads for a given virus family found in all samples analyzed together, not per sample, and had only a few reads distributed along several samples. The maximum number of reads in a single sample in these cases was 28 (Microviridae), and the second higher was 14, followed by 6 reads. We used this cutoff because we found virus families not described in humans that had this summed number of reads, so we did not consider those as relevant.
-Were there individual sequences that mapped to multiple HPV types when typing the viruses? How were these resolved? (Section 2.4)
Yes. Reads that mapped to multiple HPV references or regions were randomly assigned to one region/reference. We have included that information in the Methods section of the revised manuscript (lines 122-3).
-Although the bacterial data has previously been published, it would be useful here to at least give more data regarding the amplicon sequencing strategy (full length? Which variable regions if shorter?)
A 724 bp fragment spanning the variable regions V3 to V6 of the bacterial 16S rRNA gene was PCR-amplified using the primers 338F and 1061R. We have included this information in the Methods section of the revised manuscript (lines 129-30).
-Line 143 typo in polymerase
Done.
Figure 1 - the inclusion of the cytology on this figure is very helpful. The labels are a bit hard to read, though, so maybe indicate the types with a shape or color block above or below
The changes were done as requested.
-Figure 5 - it would be useful to see the viral data represented on this figure as well. At least at the family level.
We think that adding the virus families to this Figure will make it very busy and difficult to follow and understand. Of note, the associations between virus families and bacterial CSTs have been carried out individually and in separate parts of the manuscript (Figure 7).
-It should be noted in the discussion that the amplification, because it favors circular DNA, is an advantage for detecting some viruses (anello, papilloma,...) but may be selecting against detection of other viruses in the vagina (EBV, CMV, HSV (although some were detected, there may've been more), molluscum contagiosum, etc.).
As per the Reviewer’s request, we have added such limitation to the Discussion section of the revised manuscript (lines 326-9).
Minor points:
The title indicates the paper is about the community composition in pregnant women, but only 1/3 time points is in a pregnant woman. This should be clarified. Also this title does not truly reflect the content of the paper.
We have changed the title, suppressing the term “pregnant” as requested.
Very nice work, overall.
Thank you!
Reviewer 2 Report
This study aimed to describe the composition and dynamics of the cervical circular DNA virome (especially HPV) and the bacteriome in HIV/HPV-positive pregnant women and to address the putative interaction between the dynamics of the HPV infection, which is the main cause of cervical cancer, and the bacteriome. The human cervical microbiome is complex and its role in health and disease has just begun to be elucidated. Given that the number of samples in this study is relatively small to provide a definitive conclusion, however, I strongly agree this study is worth publishing in Viruses.
I listed several points to be considered. Please clarify in the manuscript if necessary.
Major point
(1) I am a bit confused with the term “viral load”. It can mean two things, one is relative viral load within the same sample and another is absolute viral load in each sample. Please clarify which means the authors intend to say. I am convinced broadly that the read counts could be a proxy of the viral load. In this context, the authors mean relative viral load in one single sample as the reads are shown in % but not in absolute read counts (Figure 1 and 4). In contrast in Figure 2 A and B, authors try to correlate the virus load against the clinical status, but it is still relative virus load as shown in %. I am not sure what the relative virus load means anything. Does less CD4 T-cell count relate with a high prevalence of Anelloviridae or low prevalence of HPV? LISL/HSIL is high productive infection than ASUCS? I am wondering from the data that less CD4 T-cell count correlates to less papillomavirus reads (as more Anelloviade reads) and normal/ASCUS? Or the methods authors employed in this study could provide any data which helps to speculate absolute virus load in each sample and not presented. If so, the data should be presented as these data are very informative and support the authors’ claim.
Minor points
(1) It is interesting the negative correlation is shown between CD4 T-Cell count and Anelloviridae read, suggesting a negative association with the host immune response (it is consistent with previous reports). At the same time, it is failed to show that the association between HIV viral load and Anelloviidae reads was not significant. Is it due to the small number of samples? As HIV viral load is thought to be correlated to host immunity status (negatively to CD4 T-Cell count), in general. Again please consider together with the major point (1).
(2) Please clarify whether any novel HPV types from Alphapapillomavirus or any HPV from other HPV spices (especially Gammapapillomavirus) are identified or not, in order to understand the HPV virome in cervix more deeply.
(3) In Figure 1, 30A-normal shows only Genomoviridae. However, in Figure 4, it shows HPV 6, 81, 89, 33 and 67. Is it right?
(4) The bacterial CTS could also be shown in Figure 4.
Author Response
Please find enclosed the revision to our manuscript entitled “Composite Analysis of the Virome and Bacteriome of HIV/HPV-Coinfected Pregnant Women Reveals Proxies for Immunodeficiency”, to be considered for publication in Viruses as an Article.
As you will see below, we have incorporated/modified all suggestions by the two Reviewers to our manuscript. Each individual criticism has been addressed, and all changes to the manuscript have been highlighted in yellow in the revised version of the text. All authors have read the revised version of the manuscript and agreed with this re-submission.
__________________________
REVIEWER 2
This study aimed to describe the composition and dynamics of the cervical circular DNA virome (especially HPV) and the bacteriome in HIV/HPV-positive pregnant women and to address the putative interaction between the dynamics of the HPV infection, which is the main cause of cervical cancer, and the bacteriome. The human cervical microbiome is complex and its role in health and disease has just begun to be elucidated. Given that the number of samples in this study is relatively small to provide a definitive conclusion, however, I strongly agree this study is worth publishing in Viruses.
Thank you!
I listed several points to be considered. Please clarify in the manuscript if necessary.
Major point
(1) I am a bit confused with the term “viral load”. It can mean two things, one is relative viral load within the same sample and another is absolute viral load in each sample. Please clarify which means the authors intend to say. I am convinced broadly that the read counts could be a proxy of the viral load.
It is important to mention that, despite we share the same concept with the Reviewer (that read counts may be a proxy to viral load), we used the term “viral load” carefully, as in the Discussion section (line 331), where we clearly use the term “proxy” to viral load, but not viral load itself.
In this context, the authors mean relative viral load in one single sample as the reads are shown in % but not in absolute read counts (Figure 1 and 4). In contrast in Figure 2 A and B, authors try to correlate the virus load against the clinical status, but it is still relative virus load as shown in %. I am not sure what the relative virus load means anything.
We agree with the Reviewer’s observations, and novel Spearman's correlation coefficient and Mann-Whitney U test calculations were carried out using the log10 transformed absolute number of assigned reads normalized per million reads used in the last BLASTX (see Methods). All the results remained the same (significant), so we kept the results with the relative viral abundance in the manuscript, but also included the analysis of the absolute reads in Methods and Results of the revised manuscript (see responses to Reviewer 1 above).
Does less CD4 T-cell count relate with a high prevalence of Anelloviridae or low prevalence of HPV?
We have calculated the Spearman's correlation of all virus families with the CD4 (using the relative viral count and the absolute reads normalized as described above), including Papillomaviridae, and only Anelloviridae showed significance. In other words, (less or more) Papillomaviridae are not correlated with CD4 counts. Surely, these samples were biased, since virtually all of them carried HPV infection, but even when analyzing samples where HPV reads were less represented, no significance was found.
LISL/HSIL is high productive infection than ASUCS?
We don't have enough data and number of samples to evaluate and compare the productive infection of these different cytological groups.
I am wondering from the data that less CD4 T-cell count correlates to less papillomavirus reads (as more Anelloviade reads) and normal/ASCUS? Or the methods authors employed in this study could provide any data which helps to speculate absolute virus load in each sample and not presented. If so, the data should be presented as these data are very informative and support the authors’ claim.
We think that now, by performing all analysis again using the absolute number of reads, as per the Reviewer’s suggestion, we increased the confidence of our results.
Minor points
(1) It is interesting the negative correlation is shown between CD4 T-Cell count and Anelloviridae read, suggesting a negative association with the host immune response (it is consistent with previous reports). At the same time, it is failed to show that the association between HIV viral load and Anelloviidae reads was not significant. Is it due to the small number of samples? As HIV viral load is thought to be correlated to host immunity status (negatively to CD4 T-Cell count), in general. Again please consider together with the major point (1).
As described above, we reanalyzed the data using the absolute number of reads normalized per million reads used in the final BLASTX search. The correlation of HIV viral load and Anelloviridae reamained nonsignificant, although borderline (rs = 0.445; p = 0.056). Maybe this result is due to the low number of samples used in this analysis, as pointed out by the Reviewer. Also, HIV VL is not completely correlated with the immune status of the host (i.e., there is a significant number of subjects with discordant HIV VL and CD4 values), which may additionally explain the lack of significance in our study.
(2) Please clarify whether any novel HPV types from Alphapapillomavirus or any HPV from other HPV spices (especially Gammapapillomavirus) are identified or not, in order to understand the HPV virome in cervix more deeply.
We did not find novel HPV types or genera other than Alphapapillomaviruses in our analysis.
(3) In Figure 1, 30A-normal shows only Genomoviridae. However, in Figure 4, it shows HPV 6, 81, 89, 33 and 67. Is it right?
Yes, the Reviewer is correct. This sample has a great number of reads from Genomoviridae (99.7%), and only a small fraction was assigned to Papillomaviridae (0.3%). The latter is not visible in the Figure because this percentage is too small.
(4) The bacterial CTS could also be shown in Figure 4.
It is a great suggestion. As per the Reviewer’s request, we have managed to include such data in Figures 1 and 4.
Round 2
Reviewer 2 Report
I am still not sure what authors mean by saying "viral load" shown in relative proportion in Figure 1, 2 and 4. I appreciate authors use a "proxy" of the HPV viral load, however, I am still wondering whether this means 'a proxy of the absolute HPV viral load (viral genome copy number per specimen or per cell etc. etc)" or "a relative viral load (like HPV viral load is higher than Anelloviridae load)". By showing in a proportion of reads (%), it is supposed to be the latter, I believe. In figure 1 sample 48B, out of 100% of reads, 43% was papillomavirus and 57% was Anelloviridae. The proportion seems to reflect a relative proportion of papillomavirus vs Anelloviridae, but not absolute viral load even as a proxy. In the references cited in the discussion (4 and 63) relating to this problem, it was shown that the comparison of HPV titer values (HPV copies per cell) from real-time PCR versus median sequence coverage obtained by RCA-Proton (4) or % of virus reads against total reads correlates with viral (absolute) quantity (63). Both make sense to me, however, do not seem to support the authors' case.
The point is that the data seem to suggest something like that papillomaviridae is most abundant among papillomaviridae Anelloviridae, Genomoviridae and Herpesviridae or that high-risk papillomavirus is more abundant than low-risk-HPV, but nothing about viral load (titre or copy number) itself, which seems to be claimed by authors.
>Please clarify in the text whether viral copy number/sample (viral load) is discussed or relative abundance within a single sample by showing % of viral reads.
>In the former case, please provide data (described in line 114-119?) or references to support the idea. As the way authors shown is quite novel and not established before. It must be more convincing and make this paper better.
>In the latter case, please describe it clearly and discuss what that means.
I am still sure this paper is worth publishing in Viruses, if these points are clarified.
Author Response
I am still not sure what authors mean by saying "viral load" shown in relative proportion in Figure 1, 2 and 4. I appreciate authors use a "proxy" of the HPV viral load, however, I am still wondering whether this means 'a proxy of the absolute HPV viral load (viral genome copy number per specimen or per cell etc. etc)" or "a relative viral load (like HPV viral load is higher than Anelloviridae load)". By showing in a proportion of reads (%), it is supposed to be the latter, I believe. In figure 1 sample 48B, out of 100% of reads, 43% was papillomavirus and 57% was Anelloviridae. The proportion seems to reflect a relative proportion of papillomavirus vs Anelloviridae, but not absolute viral load even as a proxy. In the references cited in the discussion (4 and 63) relating to this problem, it was shown that the comparison of HPV titer values (HPV copies per cell) from real-time PCR versus median sequence coverage obtained by RCA-Proton (4) or % of virus reads against total reads correlates with viral (absolute) quantity (63). Both make sense to me, however, do not seem to support the authors' case.
The point is that the data seem to suggest something like that papillomaviridae is most abundant among papillomaviridae Anelloviridae, Genomoviridae and Herpesviridae or that high-risk papillomavirus is more abundant than low-risk-HPV, but nothing about viral load (titre or copy number) itself, which seems to be claimed by authors.
>Please clarify in the text whether viral copy number/sample (viral load) is discussed or relative abundance within a single sample by showing % of viral reads.
>In the former case, please provide data (described in line 114-119?) or references to support the idea. As the way authors shown is quite novel and not established before. It must be more convincing and make this paper better.
>In the latter case, please describe it clearly and discuss what that means.
I am still sure this paper is worth publishing in Viruses, if these points are clarified.
We hope we have understood exactly what the Reviewer wants to be done, and we tried to comply with his/her observations as follows.
We should mention that we have not used the term “viral load” in any of those Figures mentioned (either in the Figure themselves or in their legends) or in our results in general, and we did not try to translate our results into that meaning. In other words, we did not want our data to be interpreted as viral load in any case. In the only instance we mentioned this term (in the Discussion), we tried to say that the read abundance may be a proxy for viral load (but not viral load itself). Moreover, none of the comparisons performed aimed to compare directly intra-sample virus family distribution (which would require taking into account the size of the different virus genomes), but rather a comparison of viral reads in different samples with distinct biological characteristics (e.g., CD4 T-cell counts, premalignant lesions, etc).
We agree with the Reviewer in the sense that by calculating the read abundance (%) of a given virus family against the total reads in a sample (and not only the viral reads) may better reflect the absolute virus titer of a given virus family in a sample. Therefore, we have performed the comparisons depicted in Figure 2 and re-estimated them based on the number of normalized reads per viral reads and also on the number of normalized reads per total (viral + non-viral) reads. As one can see in the attached Figure (for review purpose only), all three approaches showed similar, significant results in both comparisons done. Despite the fact that all three versions of the analyses show similar results, we have changed Figure 2 to show the later approach (normalization by total number of reads), as suggested by the Reviewer. The other Figures mentioned (1 and 4) are manly of descriptive nature, and the variation found in normalized reads per total reads between samples would preclude combining all samples in a single graph, so we kept the data as originally presented.
We truly hope the Reviewer is now satisfied with this new version, but we are ready to make additional changes as per the Reviewer´s recommendation if needed.